# Spatial variability of marine-terminating ice sheet retreat in the Puget Lowland

**Marion A. McKenzie[1*], Lauren E. Miller[1], Allison P. Lepp[1], and Regina DeWitt[2]**

[1]Department of Environmental Sciences, University of Virginia, 291 McCormick Rd., Charlottesville, VA, USA 22904 [2]Department of Physics, East Carolina University, 1000 E. 5th St., Greenville, NC, USA 27858-4353

Corresponding author: Marion McKenzie (marion.mckenzie@mines.edu)

[*]Author now affiliated with the Geology and Geological Engineering Department, Colorado School of Mines, 1105 Illinois St., Golden, CO, USA 80201

## Abstract

Understanding drivers of marine-terminating ice sheet behavior is important for constraining ice contributions to global sea-level rise. In part, the stability of marine-terminating ice is influenced by solid-Earth conditions at the grounded-ice margin. While the Cordilleran Ice Sheet (CIS) contributed significantly to global mean sea level during its final post-Last Glacial Maximum (LGM) collapse, the drivers and patterns of retreat are not well constrained. Coastal outcrops in the deglaciated Puget Lowland of Washington state - largely below sea level during glacial maxima, then uplifted above sea level via glacial isostatic adjustment (GIA) - record late Pleistocene history of the CIS. The preservation of LGM glacial and post-LGM deglacial sediments provides a unique opportunity to assess variability in marine ice-sheet behavior of the southernmost CIS. Based on paired stratigraphic and geochronological work, with a newly developed marine-reservoir correction for this region, we identify that the late-stage CIS experienced stepwise retreat into a marine environment between 15,000 and 14,000 years before present, consistent with timing of marine incursion into the region reported in earlier works. Stand-still of marine-terminating ice for at least 500 years, paired with rapid vertical landscape evolution, was followed by continued retreat of ice in a subaerial environment. These results suggest rapid rates of solid Earth uplift and topographic support (e.g., grounding-zone wedges) stabilized the ice-margin, supporting final subaerial ice retreat. This work leads to a better understanding of shallow marine and coastal ice sheet retreat and is relevant to sectors of the contemporary Antarctic and Greenland ice sheets and marine-terminating outlet glaciers.

## Plain Language Summary

Glaciers that flow directly into the ocean are capable of contributing rapidly to global sea level rise. The land surface that glaciers sit on can influence how quickly ice is lost to the ocean. Vertical movement of solid Earth, as a result of large ice losses, is capable of slowing or even halting glacial retreat in an ocean environment. Records of the interaction between land and glacial ice movement are contained in the sediment record along the coast of the Puget Lowland in Washington state. We interpret that solid Earth movement provided stability to this marine-terminating glacial ice for at least 500 years. These results are significant because this landscape is similar to parts of the Greenland Ice Sheet and the Antarctic Peninsula, indicating that the interactions seen in this area are applicable to modern glaciated regions.

## 1 Introduction

The terrain and substrate geology beneath ice sheets have the potential to affect the behavior of the overriding ice; they can influence ice flow organization, velocity, and margin positions (Weertman, 1974; Clarke et al., 1977; Clark, 1994; Whillans & van der Veen, 1997; Cuffey & Paterson, 2010; Jamieson et al., 2012; Margold et al., 2015). Coupled ice sheet and solid Earth models indicate that glacial isostatic adjustment (GIA) can stabilize marine-based grounding lines (van der Wal et al., 2015; Whitehouse et al., 2019; Wan et al., 2022) but this relationship has yet to be tested empirically. Due to the difficulty in observing subglacial conditions and solid Earth dynamics beneath modern ice sheets, we turn to the deglacial sediment record of the extinct Cordilleran Ice Sheet

(CIS) in the Puget Lowland. Specifically, we consider the marine-based southernmost part of the CIS, the Puget Lobe, which most recently advanced across the Puget Lowland during the Last Glacial Maximum (~20,000 years ago; Mullineaux et al., 1965; Easterbrook et al., 1967; Easterbrook, 1969; Porter & Swanson, 1998). The Puget Lowland records vertical land change due to tectonics and GIA from Puget Lobe advance

and retreat in the region, making it an ideal location to study influence of solid Earth on glacial ice. Margins of Greenland comprise sedimentary basins surrounded by mountains, suggesting the topographically-driven deglacial history of the Puget Lobe may be an appropriate analog to study contemporary Greenland Ice Sheet outlet glaciers (Eyles et al., 2018). Additionally, the flexural thickness of the lithosphere and mantle viscosity in

Antarctica (Whitehouse et al., 2019), is similar to that of the Puget Lowland (Nield et al., 2014). Contributing to understanding the role of topography and solid-Earth conditions on marine-based glacial ice can lead to development of a process-based model on marine-terminating retreat of modern ice sheets. The findings from this work are relevant to modern glacial systems and have implications for timing of CIS contribution to global

sea level as well as routes and timing of human migration into the Americas (Mandryk et al., 2001; Goebel et al., 2011; Lesnek et al., 2018).

## 1.1 Regional Context

      The Puget Lowland of Washington state has been glaciated at least six times

throughout the Quaternary as a result of CIS advance and retreat in the region. Glaciations occurred during marine isotope stage (MIS) 6 (~97,000 to 150,000 years ago; Easterbrook, 1969), MIS 4 (80,000 ± 20,000 years; Easterbrook et al., 1967; Easterbrook. 1969), and towards the end of MIS 2 in the Last Glacial Maximum (LGM;~17,500 cal. year BP; Mullineaux et al., 1965; Porter & Swanson, 1998). To provide context for our

work, focused on the most recent glaciation of the Puget Lowland (the Fraser glaciation), we share context of previous outcrop and lacustrine work, radiocarbon contraints, and geomorphic analyses from the region.

### 1.1.1   Pre-Fraser glaciation and Fraser glaciation deposits

85       A characteristic pre-LGM deposit in the Puget Lowland is the Lawton Clay, formed as the more southern Puget Lowland became a proglacial lake basin from ice advancement into the northern Strait of Juan de Fuca (Mullineaux et al., 1965). Southward migrating proglacial channels, active 18,000 - 20,000 years ago, formed extensive outwash plain deposits referred to as the Esperance Sands, and mark the

oncoming advance of the CIS in the Puget Lowland (Mullineaux et al., 1965; Crandell et al., 1966; Easterbrook, 1969; Clague, 1976; Booth, 1994). The final stage of ice sheet advance during late-stage MIS 2, or the Fraser glaciation, is marked by the deposition of the massive diamicton called the Vashon Till (Willis, 1898; Easterbrook, 1969; Clague, 1981; Domack, 1983; Easterbrook, 1986). Previously radiocarbon-dated wood collected

beneath the Vashon Till provides a maximum age for the timing of final ice advance to the latitude of around Seattle (47.608013°N) at ~14,500 [14]C years BP (~17,500 calendar years BP; Mullineaux et al., 1965; Porter & Swanson, 1998), although timing of maximum ice extent near Olympia, Washington (47.037872°N) is unknown and the degree of subglacial reworking and erosion of underlying strata is not well understood. The advance of the CIS near British Columbia and Southeast Alaska is thought to have occurred after 17,000 cal. years BP (Heaton & Grady, 2003; Ward et al., 2003; Lesnek et al., 2018; Dalton et al., 2020), suggesting the Puget Lobe advanced slightly before or around the same time as the western and northwestern margins of the rest of the CIS. However, rates of terminus advance are hypothesized to be between 80 and 200 meters per year (Booth, 1987; Easterbrook, 1992). Additionally, there are disagreements about synchronicity of final advance of the Puget Lobe and the more northern, westward-flowing Juan de Fuca Lobe (Bretz, 1920; Waitt and Thorson, 1983; Easterbrook, 1992).

Calculated from glacial erratics in the Cascade and Olympic Mountains, the maximum thickness of the Puget Lobe is thought to range between 300 and 1,200 meters, with the glacial lobe thickening northward (Easterbrook, 1963; 1979; Thorson, 1980). Ice surface slope (60 cm/km) and rate of basal sliding (650 m/yr), calculated from proposed ice thickness suggest at its LGM extent, suggest the Puget Lobe was comparable to modern temperate glaciers (Booth 1987; Easterbrook, 1992).

The contact between the outwash and the overlying Vashon Till rests suggests erosion, at least in part due to extensive glacial scour during till deposition (Bretz, 1913; Easterbrook, 1968; 1992). Till deformation due to high pore pressure beneath the ice (Booth, 1984) is also evidenced by preserved streamlined subglacial bedforms through deformation across the Lowlands (Booth, 1984; Goldstein, 1994; McKenzie et al., 2023). Incised channels that served as drainage pathways in the subglacial environment (Booth, 1984) are thought to have co-evolved with the subglacial streamlined bedform field (Goldstein, 1994) during LGM ice occupancy in the Puget Lowland. These geologic and geomorphic records collectively indicate highly dynamic, polythermal conditions at the ice=bed interface in association with the Vashon Till.

### *1.1.2 Deglacial and nonglacial Holocene deposits*

Recording the transition between a subglacial and glaciomarine environment is the shell-bearing Everson Glaciomarine Drift deposits, present along southern coastal outcrops in the Lowlands (Armstrong et al., 1965; Easterbrook, 1969; Powell, 1980; Thorson, 1980; Pessl et al., 1981; Domack, 1983, 1984; Dethier et al., 1995; Swanson & Caffee, 2001). The Everson Glaciomarine Drift marks intrusion of marine water into the Puget Lobe subglacial environment, primarily grounded below sea level (Thorson, 1980; Dethier et al., 1995; Demet et al., 2019). The oldest marine shells dated from the Everson Glaciomarine Drift suggest the Puget Lowland was partially deglaciated and open to marine influence by 13,470 ± 90 and 13,090 ± 90 [14]C years BP (~16,275 and 15,750

calendar years BP, respectively; Easterbook, 1992; Dethier et al., 1995; Stuiver et al., 1998; Swanson & Caffee, 2001). The lack of both sufficiently documented stratigraphic context for individual ages and a lack of marine reservoir correction (MRC) for this region, however, contribute to uncertainties in this generalized date of deglaciation in the Puget Lowland (c.f., Porter & Swanson, 1998; Swanson & Caffee, 2001). Additionally, conflicting ages from freshwater lacustrine organics on the eastern fringe of the Puget Lowland suggest ice retreat before ~13,600 [14]C years BP (~16,500 calendar years BP; Rigg & Gould, 1957; Leopold et al., 1982; Anundsen et al., 1994), and cosmogenic nuclide production rates, calculated from wide-ranging radiocarbon ages marking deglaciation indicate that retreat occurred ~15,500 years ago (Swanson & Caffee, 2001). Disagreements in timing of ice retreat remain, while emerging research suggests much of the CIS experienced Pleistocene Termination mass loss before significant climate reversals (Menounos et al., 2017; Walczak et al., 2020).

The distribution of the Everson Glaciomarine Drift has been used to suggest marine incursion inciting a rapid lift-off of grounded Puget Lobe ice (i.e., rapid transition from grounded ice to a floating ice shelf; Thorson, 1980, 1981; Waitt & Thorson, 1983; Booth, 1987; Booth et al., 2003). Synchronous retreat of the Puget Lobe and the largely westward flowing Juan de Fuca Lobe due to the decoupling of the Puget Lobe from its bed due to marine incursion has also been suggested (Easterbrook, 1992). However, major differences in deglacial stratigraphy across the Puget Lowland (Powell, 1980; Pessl et al., 1981; Domack, 1984; Demet et al., 2019), indicate variable patterns of ice-marginal retreat in time and space. The presence of grounding-zone wedges (GZWs) across the region, sedimentary deposits found at the margin of marine-terminating ice lobes, also supports the idea of stepwise retreat with periodic stability (Simkins et al., 2017; Demet et al., 2019). The development of these ice-marginal landforms were likely supported by the identified high rates of sedimentation in the region (~2.5 mm/year; Simkins et al., 2017; Simkins et al., 2018; Demet et al., 2019).

Additionally, modern elevation of marine limits in the Puget Lowland range from ~125 m above sea level in the northern San Juan islands to less than 30 m at the southern end of Whidbey Island (Thorson, 1981, 1989; Dethier et al., 1995; Kovanen & Slaymaker, 2004; Polenz et al., 2005), indicating highly variable rates of GIA across the region. Ice retreat as marked by the deposition of the Everson Glaciomarine Drift is thought to have occurred at the same time as rapid isostatic uplift between 13,600 and 11,300 [14]C yr. BP (Dethier et al., 1995). The rate of landscape emergence due to GIA in the Puget Lowland may have been as high as 10 cm/yr during early deglaciation (Dethier et al., 1995). The high rate of GIA-induced uplift during glacial retreat suggests relative sea-level fall in the Puget Lowland outpaced rapid global sea-level rise, leading to emergence of the landscape during the end of the LGM (Shugar et al., 2014; Yokoyama & Purcell, 2021). Emergence of this landscape from below to above sea level is also distinctly marked in post-glacial stratigraphy by thin subaerial deposits (e.g., fluvial

sediments and soil) overlying the glacial and glaciomarine deposits (Domack, 1984; Demet et al., 2019). Both pre-existing topography and GIA uplift could have periodically stabilized the Puget Lobe during retreat, as suggested for contemporary ice sheets (Durand et al., 2011; Favier et al., 2016; Alley et al., 2021; Robel et al., 2022), highlighting the importance of elucidating the influence of both variables on ice-sheet

behavior.

Despite the large amount of research conducted on the Puget Lobe, many of the radiocarbon ages from this region lack detailed stratigraphic context for their proposed constraints (e.g., Easterbrook, 1992 and references therein). Additionally, the continued

absence of a local MRC has left some uncertainties in the marine-based radiocarbon used in the region. Pleistocene variability in marine reservoir ages in this region (Schmuck et al., 2021) make creating a widely applicable MRC difficult, but developing an understanding of recent marine reservoir effects local to the Puget Lowland would increase reliability of marine ages. Recent advancements in technologies for radiocarbon

dating, sedimentology, and stratigraphic analyses also beg a reanalysis of Puget Lowland deposits that have not been observed using modern understandings of glaciology. Subsequently, the need to clarify spatiotemporal details of ice retreat patterns and drivers of Puget Lobe retreat persists.

**1.2 Relevance to solid Earth dynamics and modern ice sheets and glaciers**
Based on modelled evidence of GIA control on ice behavior in analogous Antarctic Peninsula glacial catchments (Nield et al., 2014; Whitehouse et al., 2019), in addition to previously identified geomorphic evidence of ice-margin stand still in the Puget Lowland (Simkins et al., 2017; Demet et al., 2019), we hypothesize that landscape

position above and below sea level, in response to loading and unloading of the solid Earth, influenced ice-margin positions and drove a punctuated retreat of the CIS during the late Pleistocene. In the central Puget Lowland, Whidbey Island spans nearly 100 kilometers in distance along the North-South direction of glacial ice movement and hosts extensive coastal bluff features (Figure 1). The outcrops, composed of glacial and

interglacial sediments, preserve details of ice advance and retreat across the formerly marine landscape, as well as landscape transitions that took place coeval with deglaciation. Except for localized tectonic deformation of surficial sediments (Sherrod et al., 2008), local LGM and subsequent deglacial deposits appear to have little post-depositional reworking (Booth & Hallet, 1993; Kovanen & Slaymaker, 2004; Eyles et al.,

2018; Demet et al., 2019; McKenzie et al., 2023).
In this work, decimeter-scale stratigraphic and sedimentological assessments are complemented by accelerator mass spectrometry radiocarbon ($^{14}C$) and optically stimulated luminescence (OSL) dating. While these two dating methods have been utilized in this region for decades (e.g., Rigg and Gould, 1957; Leopold et al., 1982;

Easterbrook, 1992; Anundsen et al., 1994; Dethier et al., 1995; Swanson and Caffee, 2001), our hypothesis of the relationship and timing of landscape emergence in relation to ice retreat and periodic stabilization of ice retreat has not been directly assessed. Additionally, with this work, we develop a modern and regional MRC that is applied to the dates for marine deglaciation. Therefore, the application of advances in geochronology paired with a high-resolution stratigraphic assessment of Whidbey Island is a novel approach to elucidating the ice retreat and land emergence across the region.

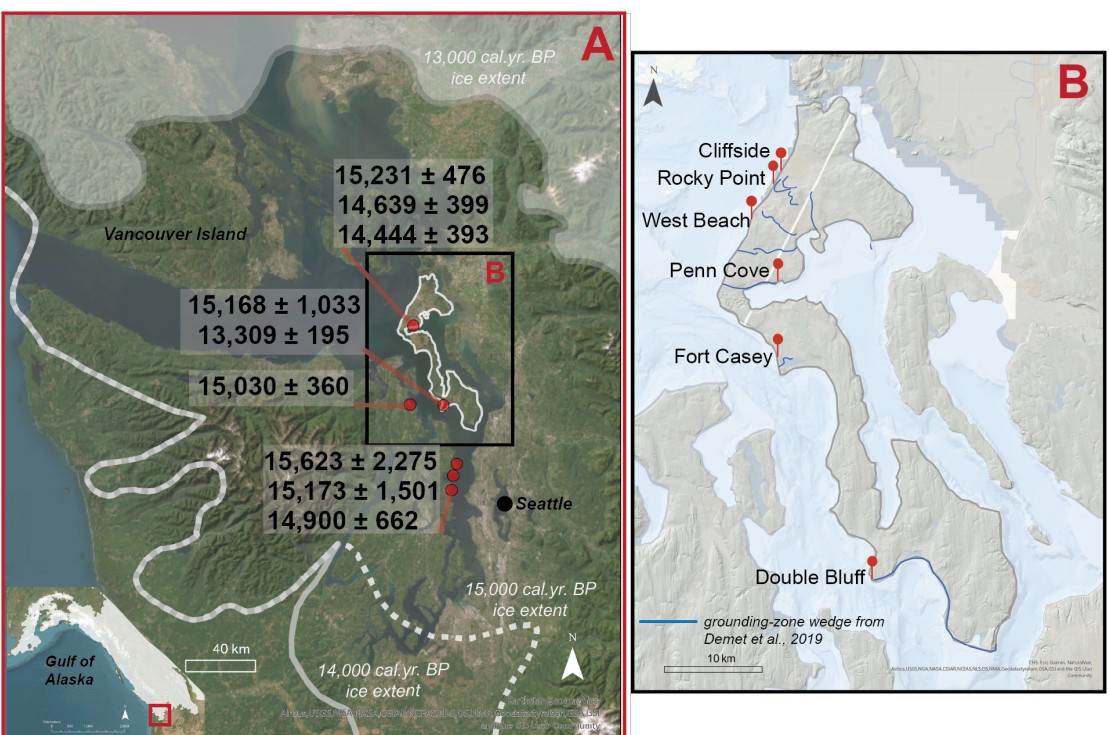

**Figure 1.** A) Regional map showcasing maximum extent of the Cordilleran Ice Sheet (CIS) at 15 kya, 14 kya, and 13 kya cal. yr. BP (data synthesized by Henry Haro from Ehlers et al., 2010). This map has been adapted from Earthstar Geographics satellite imagery. Glaciomarine shell radiocarbon dates from the literature (red dots) were recalibrated using Marine20 and new marine reservoir correction are listed in cal. yr. BP. Only glaciomarine shell ages with available metadata and error standards relevant to work presented here were included in the analysis (e.g., Easterbrook, 1992; Dethier et al., 1995; Swanson and Caffee, 2001). Information about ages and recalibration conducted in this work may be found in Table 1. B) Whidbey Island inset map with sites labelled south to north. Grounding zone wedges (GZWs) identified and inferred from Demet et al., 2019 are mapped in blue. Streaming of the bed in visible around the margins of the inset and on the southwest side of Whidbey Island (outlined in gray). This map has been adapted from data provided by Esri, Garmin, NaturalVue, Airbus, USGS, NGS, NASA, CGIAR, NCEAS, NLS, OS, NMA, Geostastyrelsen, GSA, GSI, and the GIS user interface.

## 2 Materials and Methods

### 2.1 Sedimentology and stratigraphy

Sediment samples were collected from Whidbey Island outcrops a) Double Bluff, b) Fort Casey, c) Penn Cove, d) West Beach, and e) Cliffside at 10-cm intervals (Figure 1B; Table S1) with additional subsamples collected from units with laminations, lenses, or rip up clasts. Sites were determined based on accessibility from the beach front. If some outcrop units were pinched out while others continued, these multiple-location data collection were considered as one site (indicated by white dots on the stratigraphic columns in Figure 2). However, if continuity between units was not visible or accessible across a single beach front, these sites were separated into "1" and "2" such as in the case of Fort Casey and West Beach. Thin (~ <0.5cm thick) horizontally continuous layers are referred to as laminations, while less continuous layers that pinch out are referred to as lenses (e.g., Figure S1). Over 300 discrete bulk sediment samples were analyzed at the University of Virginia for grain size and magnetic susceptibility (MS). An additional 15 peat, wood, and marine shell samples were excavated for radiocarbon dating. Grain size analyses were conducted via a BetterSize S3 Plus Particle analyzer on sample matrix material (material ≤ 3 mm in diameter) and MS measurements were collected with a Bartington MS2 magnetic susceptibility meter. MS values provide information about amount and size of magnetic grains in each sample, elucidating continuity and source of biogenic and lithogenic deposits (Thompson and Oldfield, 1986; Verosub and Roberts, 1995; Rosenbaum, 2005; Hatfield et al., 2017; Reilly et al., 2019). Grain sizes from the BetterSize S3 Particle analyzer are categorized as silt/clay (0.06 – 63 µm), very fine sand (63 - 125 µm), or fine sand (125 – 250 µm) (Wentworth, 1922; Folk & Ward, 1957). Results of the Whidbey Island stratigraphy are presented according to latitudinal location, starting with the southernmost site, Double Bluff, followed by the Fort Casey Sites, Penn Cove, West Beach sites, and ending with the northernmost Cliffside and Rocky Point sites.

### 2.2 Accelerator Mass Spectrometry radiocarbon analysis

Assuming a constant cosmically produced $^{14}C$ to $^{12}C$ ratio, the variation in this ratio can be used to determine the amount of time since the death of formerly living specimens. Samples were run at the National Oceanographic Sciences Accelerator Mass Spectrometry (NOSAMS) Laboratory at Woods Hole Oceanographic Institute. The unprocessed wood material underwent a series of six to eight acid-base-acid leaches to remove contamination and inorganic carbon prior to combustion. The carbonate shell samples underwent carbonate hydrolysis and resulting carbon combustion reacted with Fe catalyst along vacuum-sealed lines to produce graphite (NOSAMS, 2023). Resulting graphite pellets were pressed into targets and analyzed by accelerator mass spectrometry in addition to standard and processing blanks (Roberts et al., 2019). The AMS measurements determined the ratio of $^{14}C$ to $^{12}C$ in each of the pellets, which was then

used to calculate the radiocarbon age using the Libby $^{14}$C half-life of 5,568 years (Stuiver and Polach, 1977; Stuiver, 1980).

Conversion of radiocarbon years to calendar years BP was conducted using the Int20 curve for terrestrial carbon samples and the Marine20 curve for marine shell samples using the Calib 8.2 interface (Heaton et al., 2020). Marine20 is the baseline marine curve used for Calib 8.2 and is the most up-to-date, internationally agreed marine radiocarbon age calibration curve for non-polar global-average marine records (Heaton et al., 2020). A modern marine reservoir correction was calculated in Calib 8.2 and applied to all carbonate shell samples, both newly presented in this work and to previously published radiocarbon ages, using contemporary shells with known pre-1955 (i.e., prior to nuclear bomb testing) collected dates from the Burke Museum in Seattle, Washington. The modern (pre-1955) shells from the Burke Institute were live-collected between 1911 and 1931. Species of the modern bivalces include *Modiolus rectus, Musculus niger, Cardita ventricas, Macoma carlottensis, Mya arenaria,* and *Macoma nasuta*. The radiocarbon ages calculated from these specimens range from $815 \pm 15$ to $925 \pm 20$ $^{14}$C years (Table S2). Utilizing the marine reservoir correction curve developed by Calib 8.2, an average marine reservoir correction for this region is 264 $^{14}$C years, which was applied to other marine shells from this work and previously published work (Figure 1). While there is a narrow range of marine reservoir effects between 211 and 318 $^{14}$C years, a species-specific effect was not observed. It is also important to note that due to known Pleistocene variability in marine-reservoir ages in this region (Schmuck et al., 2021), this marine-reservoir correction is most accurate for marine organisms aged between 1911 and 1931. However, while these limitations exist, this new reservoir age is highly localized to the Puget Lowland and was therefore extended for use in older collected marine shells.

**2.3 Optically stimulated luminescence**

In depositional environments, minerals are exposed to radiation from in situ uranium (Ur), thorium (Th), potassium (K), and cosmic rays (Rhodes, 2011; Duller, 2015). Incoming radiation excites electrons which are trapped in structure deformities of quartz and feldspar grains (Rhodes, 2011). When exposed to sunlight, electrons are released from the traps. In returning to their original states, they emit luminescence and the mineral is reset. Upon burial, trapped electrons re-accumulate, and the amount is proportional to the burial time and the radiation exposure, termed "dose". The rate of irridation, the "dose rate," can be calculated from the cosmic flux as well as the U, Th, and $^{40}$K concentrations of the surrounding sediments. The OSL signal is proportional to the dose and can be measured by exposing the mineral to light in a controlled setting. An age since burial can be determined by dividing the dose by the dose rate (Tables S3, S4, S5).

Materials from glacial environments present challenges due to the potential of the OSL signal not being fully reset between transport and deposition (Wallinga and

Cunningham, 2015). Additionally, extensive overburden pressure from glacial ice has the potential to partially or completely reset OSL signatures, which could provide large error to the final OSL stage (Bateman et al., 2012; King et al., 2014). Subglacial environments, especially those under ice streams, have a presence of significant meltwater which can saturate sediment pore space and influence quartz and feldspar exposure to radiation at the time of and for an extended period of time after deposition (Wallinga and Cunningham, 2015; Duller, 2013).

While a detailed description of the OSL procedure can be found in supplement text (Text S1), a summary is provided here. In order to avoid pre-mature bleaching of samples, they were collected before sunrise or after sunset, only exposed to low energy red light, and wrapped in dark black plastic before being transported to East Carolina University (ECU) for preparation and processing. Samples were prepared for OSL analysis under dark-room conditions using standard procedures to extract 63-212 μm quartz. Due to feldspar contamination, a post-IR blue SAR procedure was used to measure the quartz equivalent dose (Murray and Wintle, 2000; Wallinga et al., 2002; Wintle and Murray, 2006).

Bulk sediment was collected from outcrops for high-resolution gamma spectrometry measurements and stored for at least 4 weeks prior to measurement. OSL samples were taken at unit boundaries, while dose rate samples were only taken from the same unit as the OSL samples. Therefore, the gamma dose rates reflect the sample unit only and contain no information about adjacent, underlying, or overlying units. Uranium concentrations determined from $^{234}$Th were all significantly higher than concentrations determined from $^{214}$Pb and $^{214}$Bi. We assumed that $^{234}$U was leached out of the sample due to in situ water presence.

The sample ages, calculated in calendar years, were calculated by dividing the dose by the dose-rate (Tables S3, S4, S5). For samples with feldspar contamination that showed fading, the ages were corrected as suggested by Auclair et al., (2003). While $^{14}$C ages are reported in kilo years ago (kya) calendar year BP (1955), all OSL ages are reported in kya based on the date of collection (2020). OSL ages in kya can be directly compared to kya cal. BP by subtracting 72 years from the OSL age. Cases of large overdispersion (> 20%) indicate mixing of grains or non-uniform resetting of the samples. Possible reasons could be from bioturbation, mixing with light-exposed surface grains during collection, or incomplete resetting during deposition. In results, we cannot attribute much relevance to overdispersed samples.

**3 Results**

We will be moving through results from the southernmost to the northernmost site. Numerical schemes to describe units at each site are independent and do not correlate between sites (Figure 2). Stratigraphic columns were developed to represent our interpretation of physical data present at several locations across these sites (Figure 2).

### 3.1 Double Bluff

The stratigraphically lowermost unit visible at Double Bluff, Unit 4, is a visually well-sorted sand with sparse rounded gravel lenses. Unit 4 is normally graded with clasts ranging from granule to pebbles with a consistent horizonal long-axis orientation and occasional silt rip ups from nonvisible underlying units. A gradational boundary leads into the overlying sandy silt and fine clayey silt of Unit 3. This unit contains wavy laminations and woody debris dated to be 46.7+ thousand years (kya) cal. BP (i.e., "radiocarbon dead"; Table 1 NOSAMS Receipt #171378). Unit 3 generally fines upwards but with variable matrix grain size modes from 10-500 μm (Figure 2). Unit 2 is composed of massive diamicton with a clay and fine-silt matrix, marked by a matrix grain size mode of 8 μm and a mix of angular and rounded granule to cobble-sized clasts without a preferred long-axis orientation. There is a gradational contact between Unit 2 and Unit 1. Unit 1 consists of diamicton with a matrix varying between sandy silt and silty sand with woody debris dated to 48.0+ kya cal. BP in age (i.e., "radiocarbon dead"; Table 1 NOSAMS Receipt #176245) and clasts that are predominantly aligned parallel to bedding and evidence of soft-sediment deformation. This uppermost unit has interbedded silt and clay, as well as marine shells in the upper 50 cm of silt that were inaccessible for sampling. MS values in Unit 3 are distinctly lower than the other units (Figure S2).

### 3.2 Fort Casey

The lowermost visible unit, Unit 3, at Fort Casey Site 1 consists of massive diamicton with a fine-silt and clay matrix and randomly oriented pebble to cobble-sized angular and rounded clasts. Interbedded with the massive diamicton are discrete gravel and sand laminations at the base of Unit 3 and silt and clay laminations with rip ups and woody debris toward the top of Unit 3. Unit 2 consists of fine sand to pebble-size clasts in a sandy silt matrix with vertically oriented and reverse-graded angular clasts. Unit 2 has a remarkably consistent matrix grain size throughout the unit. An OSL sample from this unit (Table 2, Sample #1) could not be dated reliably due to extremely low signals. This unit also contains sand and silt lenses with mud and plant rip ups (Figure 2). A gradational boundary leads to Unit 1, which is massive diamicton similar to Unit 3 but with a matrix distinctly lighter in color.

At Fort Casey Site 2, the lower visible unit, Unit 5, contains interbedded clay and sand with reverse grading (Figure 2). Unit 4, in which no samples were collected, consists of diamicton with concentrated granule to pebble lenses and clay and silt lenses, as well as evidence of soft-sediment deformation. Unit 3 is a massive clay, followed by the Unit 2 layer of silt about 20 cm thick, continuous across an irregular, undulating, and most likely erosional contact. OSL dates at the top of Unit 2 and base of Unit 1 were found to be $40.8 \pm 8.2$ and $56.6 \pm 15.5$ kya (Table 2 Samples #3, 2). The overlying Unit 1 is a diamicton with very fine sand to cobble sized angular and rounded clasts. Normal

grading is present in the matrix of Unit 1 with fractured (i.e., seemingly crushed) granite clasts.

### 3.3 Penn Cove

The lowest visible unit at this site, Unit 5, comprises a reverse-graded diamicton with a coarsening upward sand matrix and rounded granules and pebbles (Figure 2). Following a sharp boundary with Unit 5, Unit 4 consists of silt and sand laminations with cross-bedded sands near the top. Unit 4 deposits were OSL dated to ages 56.6 ± 4.1 and 44.4 ± 2.8 kya (Table 2 Samples #10, 11), where the sample with older age showed slightly large overdispersion (> 20%) which we attribute to incomplete resetting. The grain size modes for Unit 4 matrix are predominantly between 500-700 μm (Figure 2). An erosional boundary at the top of Unit 4 leads to the massive clayey silt diamicton of Unit 3 with rounded fine- to cobble-size clasts and occasional sandy silt and silt lenses. A gradational boundary separates Units 3 and 2, which is a massive clay diamicton with rounded fine sand to cobble grains and articulated shells. Six shells from Unit 2 were radiocarbon dated with ages spanning 14.8 ± 0.3 to 14.1 ± 0.3 kya cal. BP (Table 1 NOSAMS Receipt #176239-176242, 171380, 171381). Unit 2 also contains sand lenses and wood fragments. Unit 2 has a sharp contact with Unit 1, which consists of normally graded gravel with rounded and angular small to large pebbles with no predominant long-axis orientation. A mode of clay-sized grains is visible in Units 2 and 3 but is not visible in Unit 1 (Figure 2).

### 3.4 West Beach

At West Beach Site 1, the lowest unit, Unit 5, consists of matrix-supported diamicton with randomly orientated clasts and two matrix grain size modes at 8 and 20 μm (Figure 2). This unit has a sandy-silt lamination that interrupts the diamicton. The diamicton above the silty-sand lamination, however, contains highly irregular dips and soft-sediment deformation. Unit 5 has a gradational boundary with Unit 4 – a light clay layer deposited on a laterally irregular surface, marked by normal-grading, or fining upward (Figure 2). Unit 3 consists of a thick, 0.25-m clast-supported gravel layer with poorly sorted fine sand to cobble size clasts oriented parallel to the depositional bed. A sharp, horizontally regular contact occurs between Unit 3 to the 0.75 m-thick, well-sorted sand of Unit 2 with OSL ages of 6.2 ± 0.6 and 4.1 ± 1.8 kya (Table 2 Samples #5, 4), however, these ages experienced large overdispersion (> 20%). Unit 2 has a gradational contact with Unit 1, which is a modern soil on top of a basal shell hash dating between 0.62 ± 0.1 and 0.39 ± 0.1 kya cal. BP (Table 1 NOSAMS Receipt #176236-176238). MS values are similar throughout Units 5, 4, 2, and 1, but decrease in Unit 3 (Figure S2).

At the base of West Beach Site 2 are cross-bedded and coarse sand laminations. OSL dates from the lowermost sand in Unit 8 are dated to 31.3 ± 2.7 and 38.1 ± 9.7 kya (Table 2 Sample #7, 6), with overlying Unit 7 sediments OSL dated between 30.7 ± 2.5

and 29.2 ± 4.6 kya (Table 2 Sample #8, 9), however, these ages experienced large
overdispersion (> 20%). A gradational contact leads into Unit 7, consisting of silt and
clay with radiocarbon-dead woody debris (48.0+ kya cal. BP; NOSAMS Receipt
#176243). Unit 6 consists of sand with wavy bedding and silt laminations. No samples
were collected from Units 5 and 4, consisting of a peat layer and a unit of sand and silt
laminations, respectively. The Unit 3 diamicton matrix coarsens upwards and this unit
has many grain size modes between 5 and 70 μm (Figure 2). Unit 2 consists of diamicton
with a fine sand matrix and clasts as large as pebbles and is not spatially continuous
throughout the site. A gradational boundary leads into the 0.5 m-thick layer of Unit 1,
consisting of predominantly of silt.

## 3.5 Rocky Point, Cliffside

The lowest visible unit at Cliffside, Unit 6, consists of fine sand to cobble-sized rounded
clasts. This massive diamicton has no preferential orientation for clast long axes. The
matrix changes from clay to sand and includes sediment deformation beneath clasts
(Figure 2). Unit 6 gradationally transitions to Unit 5, which is a normally graded, fine
sand to cobble-size clast diamicton. Unit 5 is normally graded gravel lenses containing
clasts with consistent horizontal long-axis orientation. Unit 5 gradually transitions into
the granule and sand layer of Unit 4, which includes sand and silt lenses within gravel-
rich and wavy laminations. Unit 3 intrudes into Unit 4 and consists of a massive
diamicton with rounded, cobble-sized clasts. The matrix of Unit 3 has two grain size
modes at 5 and 20 μm (Figure 2). Two of the lowerunit samples for Cliffside Unit 3 were
taken from the more southern Rocky Point site as the identified Unit 3 is continuous
throughout both sites. Unit 3 gradually transitions into Unit 2, which is a laterally
discontinuous light clay unit with silt layers. Unit 1 is comprised of mostly rounded,
normally graded crushed material with fine to large cobble size clasts.

**Table 1.** Radiocarbon sample descriptions and data. Gray rows indicate previously
published radiocarbon data that have been recalibrated using Marine20 and our MRC.

| Sample location (site, unit) | Laboratory number | Type | Age ± error (RCY) | Applied MRC | Age ± 2σ (cal year BP) | Lat (°N) | Long (°W) | Source | NOSAMS Receipt # |
|---|---|---|---|---|---|---|---|---|---|
| West Beach Site 1, Unit 1 | WB S1 RCD1 s.h. base U6 | marine shell | 1290 ± 20 | 278 ± 35 | 590 ± 2 | 48.3 | 122.73 | this work | 176236 |
| West Beach Site 1, Unit 2 | WB S1 RCD1 s.h. base U6 clam | marine shell | 1210 ± 25 | 278 ± 35 | 385 ± 130 | 48.3 | 122.73 | this work | 176237 |
| West Beach Site 1, Unit 3 | WB S1 U6 RCD2 | marine shell | 1450 ± 15 | 236 ± 30 | 615 ± 113 | 48.3 | 122.73 | this work | 176238 |
| Penn Cove, Unit 3 | PC S3 U3 RCD3 | glaciomarine shell | 13200 ± 75 | 278 ± 35 | 14579 ± 371 | 48.24 | 122.69 | this work | 176239 |
| Penn Cove, Unit 2 | PC S3 U4 RCD5 | glaciomarine shell | 13000 ± 75 | 271 ± 35 | 14317 ± 363 | 48.24 | 122.69 | this work | 176240 |
| Penn Cove, Unit 2 | PC S3 U4 RCD1 a.s. | glaciomarine shell | 13250 ± 75 | 264 ± 36 | 14673 ± 366 | 48.24 | 122.69 | this work | 176241 |
| Penn Cove, Unit 2-3 transition | PC S3-4 RCD2 | glaciomarine shell | 12900 ± 55 | 264 ± 36 | 14103 ± 294 | 48.24 | 122.69 | this work | 171380 |
| Penn Cove, Unit 2 | PC S3 RCD4 | glaciomarine shell | 13200 ± 55 | 264 ± 36 | 14609 ± 334 | 48.24 | 122.69 | this work | 171381 |
| Penn Cove, Unit 2 | PC S3 U4 RCD3 | glaciomarine shell | 13300 ± 75 | 216 ± 30 | 14833 ± 343 | 48.24 | 122.69 | this work | 176242 |
| West Beach Site 2, Unit 7 | WB S2 U1 RCD1 | reworked wood | > 48000 | N/A | N/A | 48.3 | 122.73 | this work | 176243 |
| Double Bluff, Unit 4 | DB S3 RCD1 U4 | reworked wood | > 46700 | N/A | N/A | 47.97 | 122.55 | this work | 171378 |
| Double Bluff, Unit 1 | DB S5 RCD1 U7 | reworked wood | > 48000 | N/A | N/A | 47.97 | 122.55 | this work | 176245 |
| Oak Harbor | Beta-1319 | glaciomarine shell | 13650 ± 350 | 264 ± 36 | 15168 ± 1033 | 47.97 | 122.53 | Dethier et al., 1995 | N/A |
| Oak Harbor | Beta-1716 | glaciomarine shell | 13600 ± 150 | 264 ± 36 | 15231 ± 476 | 48.23 | 122.70 | Dethier et al., 1995 | N/A |
| Basalt Point | AA-10077 | glaciomarine shell | 13470 ± 90 | 264 ± 36 | 15030 ± 360 | 47.97 | 122.72 | Swason and Caffee, 2001 | N/A |
| Double Bluff | QL-4608 | glaciomarine shell | 12260 ± 60 | 264 ± 36 | 13309 ± 195 | 47.97 | 122.53 | Swason and Caffee, 2001 | N/A |
| Penn Cove | PC-01 (UWAMS) | glaciomarine shell | 13230 ± 90 | 264 ± 36 | 14639 ± 399 | 48.23 | 122.70 | Swason and Caffee, 2001 | N/A |
| Penn Cove | PC-02 (UWAMS) | glaciomarine shell | 13090 ± 90 | 264 ± 36 | 14444 ± 393 | 48.23 | 122.70 | Swason and Caffee, 2001 | N/A |
| Bainbridge Island | unknown | glaciomarine shell | 14000 ± 900 | 264 ± 36 | 15623 ± 2275 | approx. 47.69 | approx. 122.50 | Easterbrook, 1992 | N/A |
| Bainbridge Island | unknown | glaciomarine shell | 13650 ± 550 | 264 ± 36 | 15173 ± 1501 | approx. 47.72 | approx. 122.51 | Easterbrook, 1992 | N/A |
| Suquamish | unknown | glaciomarine shell | 13430 ± 200 | 264 ± 36 | 14900 ± 662 | approx. 47.75 | approx. 122.47 | Easterbrook, 1992 | N/A |

**Table 2.** OSL age data with overdispersion percentages and total dose rate values. Final
sample ages are bolded. To directly compare OSL and [14]C ages, it would be necessary to
subtract 72 years from the OSL ages. This correction is considerably smaller than the

uncertainty of the ages and can therefore be neglected. Additional dose and dose rate data may be found in Tables S4 and S5.

| Sample | Sample # | Age unfaded (ka) | | Age after fading (ka) | | overdispersion (%) | Total dose rate (mGy/a) | | |
|---|---|---|---|---|---|---|---|---|---|
| FCS1-OSL1 | 1 | >9.3 ± | 2.30 | | | 78 | 1.30 | ± | 0.09 |
| | | | | | | | | | |
| FCS2-OSL1 | 2 | 41.18 ± | 3.76 | 56.6 ± | 15.5 | 20 | 2.26 | ± | 0.11 |
| FCS2-OSL2 | 3 | 32.48 ± | 2.65 | 40.8 ± | 8.2 | 11 | 2.13 | ± | 0.10 |
| | | | | | | | | | |
| WBS1-OSL1 | 4 | 3.40 ± | 0.46 | 4.10 ± | 1.78 | 32 | 2.20 | ± | 0.22 |
| WBS1-OSL2 | 5 | 6.24 ± | 0.59 | | | 24 | 1.90 | ± | 0.14 |
| WBS2-OSL1 | 6 | 27.22 ± | 3.39 | 38.05 ± | 9.65 | 29 | 2.37 | ± | 0.24 |
| WBS2-OSL2 | 7 | 31.27 ± | 2.65 | | | 19 | 2.27 | ± | 0.14 |
| WBS3-OSL1 | 8 | 30.71 ± | 2.50 | | | 26 | 2.23 | ± | 0.08 |
| WBS3-OSL2 | 9 | 22.64 ± | 2.19 | 29.18 ± | 4.63 | 33 | 2.47 | ± | 0.11 |
| | | | | | | | | | |
| PCS2-OSL1 | 10 | 36.80 ± | 3.29 | 56.60 ± | 4.14 | 25 | 2.05 | ± | 0.10 |
| PCS2-OSL2 | 11 | 44.39 ± | 2.82 | | | 7 | 2.10 | ± | 0.10 |


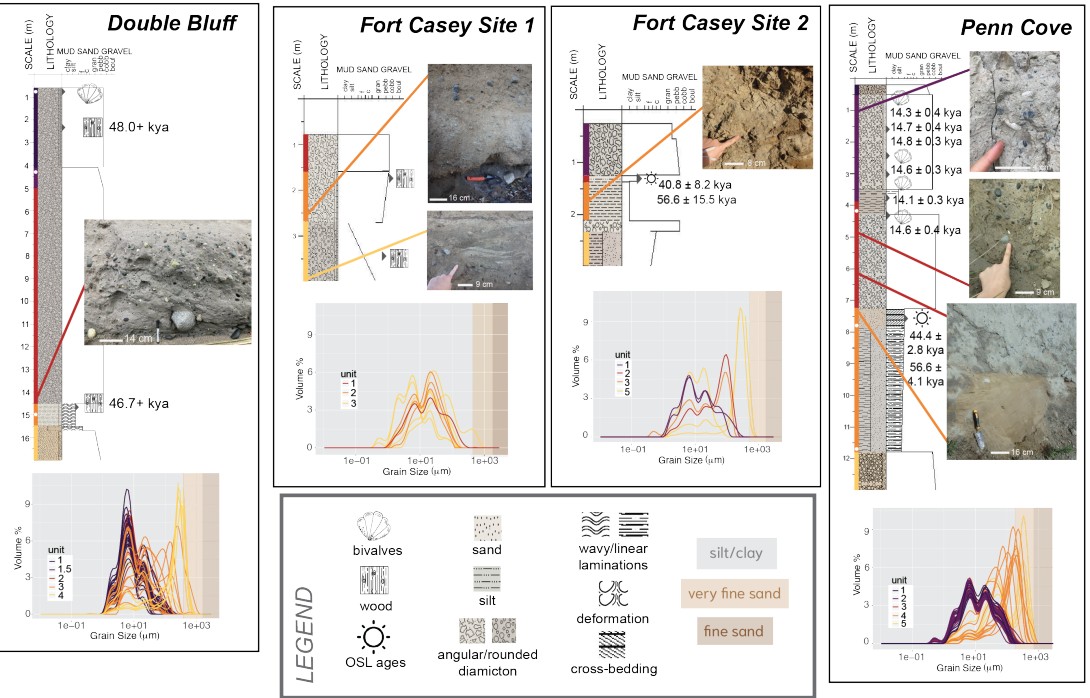

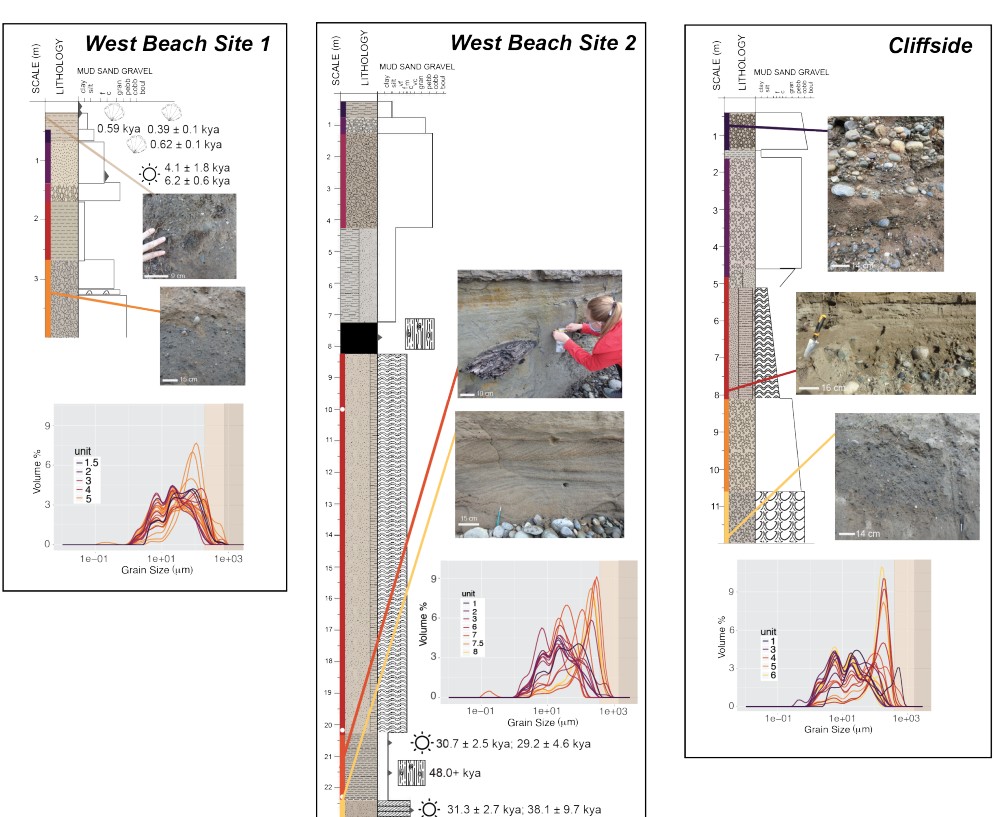

**Figure 2.** Outcrop sites from south to north: Double Bluff, Fort Casey 1, Fort Casey 2, Penn Cove, West Beach Site 1, West Beach Site 2, and Cliffside represented by stratigraphic column with collected radiocarbon and OSL and grain size data. Icons

indicate where shells or wood were present in the stratigraphy. Not all occurences of wood or shells were radiocarbon dated. The white dots on the stratigraphic columns indicate the end of one visible region and start of a new location where visible units were mapped. Colors alongside stratigraphic units indicate grain size graph correlations and are not correlated between sites – each site is independently considered. Background

colors on the grain size graphs indicate transitions in grain size. From left to right, gray is clay/silt, light brown is very fine sand, dark brown is fine sand. Variations in sampling resolution are a function of accessibility to outcrops from the beach front. Some units were more accessible for sampling than others.

## 4 Interpretation and Discussion


We use the sedimentological units described in Section 3 to establish a facies model that encompasses glaciomarine and coastal sedimentary processes and depositional environments (i.e., emergent or submergent landscape). Aided by geochronological constraints, this facies model is applied to the stratigraphic sequences observed at each

site to construct a regional history of ice behavior and landscape evolution before, during, and following the LGM (Figure 3).

### 4.1 Facies interpretation

Structureless diamicton with randomly oriented clasts of variable size, roundness,

lithology, and a range in matrix size are classified as **till,** or sediments deposited directly by glaciers in the subglacial environment (Boulton and Deynoux, 1981; Evans, 2006; Sengupta, 2017). Some biological material may be incorporated into till in the form of broken shells or woody fragments. This reworked biogenic material may be incorporated into the ice as it moves across the landscape, therefore radiocarbon ages of biogenic

material will be older than glacial occupation. These characteristics are consistent with glaciomarine tills described offshore of West Antarctica (e.g., Kirschner et al., 2012; Prothro et al., 2018; Smith et al., 2019) and western Greenland (Sheldon et al., 2016; O'Regan et al., 2021), as well as tills deposited by the relict British-Irish Ice Sheet (Evans and Thompson, 2010). Lower boundaries of till units are often characterized by erosional

contacts, reflecting glacial advance and erosion of pre-existing substrate, and may contain rip up clasts from underlying units. Due to similarities in facies formerly identified as tills and some presence of reworked woody fragments, units classified as (local) LGM till (i.e., Vashon Till) in the Puget Lowland include Unit 2 from Double Bluff, Unit 3 at Fort Casey Site 1, Unit 1 from Fort Casey Site 2, Unit 3 from Penn Cove, Unit 5 from West

Beach Site 1, and Unit 6 from Cliffside (Figures 2, 3A). Little post-depositional erosion or reworking of this glacial material is consistent with previous work identifying tills in the region (Booth & Hallet, 1993; Kovanen & Slaymaker, 2004; Eyles et al., 2018; Demet et al., 2019).

**Glacial outwash** is characterized as diamicton with a range of well-rounded and

some angular clasts with parallel-to-bedding clast orientation that suggests sediment transport via proglacial meltwater from an upstream source of glacial ice (Boulton and

Deynoux, 1981). This facies may indicate deposition in a subaerial or subaqueous environment, but importantly, clast orientation distinguishes proglacial outwash from till (Hagg, 2022). The deposits may also exhibit normal grading and/or sedimentary

structures indicative of soft-sediment deformation (e.g., loading structures, flame structures, sediment deformation beneath clasts; Boulton and Deynoux, 1981). Glacial outwash recorded in British Columbia (Clague, 1975) and the forefield of Mýrdalsjökull ice cap in Iceland (Kjær et al., 2004) feature similar structures seen in several units among our Puget Lowland outcrop sites. Using the defined classification of glacial

outwash, Units 1 and 2 from Fort Casey Site 1, Units 4 and 5 from Fort Casey Site 2, Units 1 and 5 from Penn Cove, and Units 1 and 3 from Cliffside are interpreted as glacial outwash deposits (Figures 2, 3A).

A third diamicton, structurally similar to those interpreted as till yet containing articulated and/or broken marine shells, occasional absence of fines from winnowing of

fine-matrix material, and sedimentary structures such as wavy laminations, is interpreted as **glaciomarine deposits,** composed of both glacial and pelagic sediments that accumulate on the ocean floor seaward of the ice margin. The winnowing of fine-grained material may be a product of tidal currents in a submarine or coastal setting (Smith et al., 2019). Such pelagic sediments have been sampled from a geographically-diverse

population of sediment cores from deglaciated continental margins (e.g., Anderson et al., 1980; Prothro et al., 2018; Smith et al., 2019), although preservation of shells and other carbonate-based materials are less common in Antarctic glaciomarine sediments. Glaciomarine deposits are also identified in coastal outcrop deposits of northern Svalbard with similar characteristics (Alexanderson et al., 2018). Both Unit 1 from Double Bluff

and Unit 2 from Penn Cove are consistent with these classifications and closely resemble the structure and composition of the glaciomarine deposits identified on deglaciated continental margins (Figures 2, 3A; Anderson et al., 1980; Prothro et al., 2018). At sites Double Bluff and Penn Cove, this facies (a.k.a. Everson Glaciomarine Drift) overlays till, indicating ice marginal retreat into a marine setting with sand-rich deposits recording

removal of fines by bottom currents. Conversely, till that stratigraphically transitions upsection into cross-bedded sands or gravels with parallel-to-bed oriented clasts, occassional wavy laminations, and are barren of marine shells indicate retreat into a subaerial environment, as is observed proximal to the Mýrdalsjökull ice cap in Iceland (Kjær et al., 2004). Unit 3 from West Beach Site 1 and Unit 5 from Cliffside record such

evidence of **subaerial glacial retreat** both meet these classifications (Figures 2, 3A).

Facies transitions where grain sizes coarsen-upward (i.e., reverse grading) and changes in MS values can be associated with marine to coastal environment transition (e.g., Komar, 1977; McCabe, 1986; Sengupta, 2017) as a result of **landscape emergence.** Regardless of the process(es) explaining the observed grain coarsening, which may

include tectonic activity, glacial isostatic response, or a combination of these factors, we would expect such processes to be marked by facies transitions along the coast. In the

Puget Lowland, emergence above sea level has been recorded in the stratigraphy by thin subaerial deposits (e.g., fluvial sediments and soil) overlying the glacial and glaciomarine deposits (Domack, 1984; Demet et al., 2019). Tectonic activity as a cause for coarsening-
upward trends in the stratigraphy can be rejected because the till and sedimentary structures, like cross-bedding, have been preserved in the record. Coarsening-upward grain sizes seen in the transition from finer marine sediments to coastal deposits have been identified in coastal outcrops in northern Ireland (McCabe, 1986) and Svalbard (Alexanderson et al., 2018) and are interpreted to indicate relative sea level fall. While
glacial isostatic rebound is not responsible for the shallowing-upward of Svalbard facies (Alexanderson et al., 2018), the facies and coarsening material identified between Units 3 and 2 at Fort Casey Site 2, transition from Unit 5 laminated silt to Unit 4 cross-bedded sand at Penn Cove, and coarsening of grain size with peaks and MS across Units 7 and 6 at West Beach Site 2 could be connected to land emergence events (Figures 2, 3A).

Facies transitions where grain-sizes fine upward (i.e., normal grading), correspond with increases or decreases in MS, and are accompanied by the appearance of marine shells or reworked wood or terrestrial carbon are associated with **landscape submergence** (Komar, 1977; Sengupta, 2017). A specific example from the sedimentological record that marks the transition from a subaerial to a submarine
environment is from seismic profiles and regional stratigraphic data in the southwestern Pacific in South Island, New Zealand (Carter et al., 1986). The fining of material between Unit 4 sand deposits to Unit 3 silts, both with reworked, radiocarbon dead wood at Double Bluff, introduction of shells to the fining material between Units 2 and 1 at West Beach Site 1, and fining of grain size across the Unit 2 and 1 boundary at West Beach
Site 2 are all interpreted as a transition to a submarine setting (Figures 2, 3A).

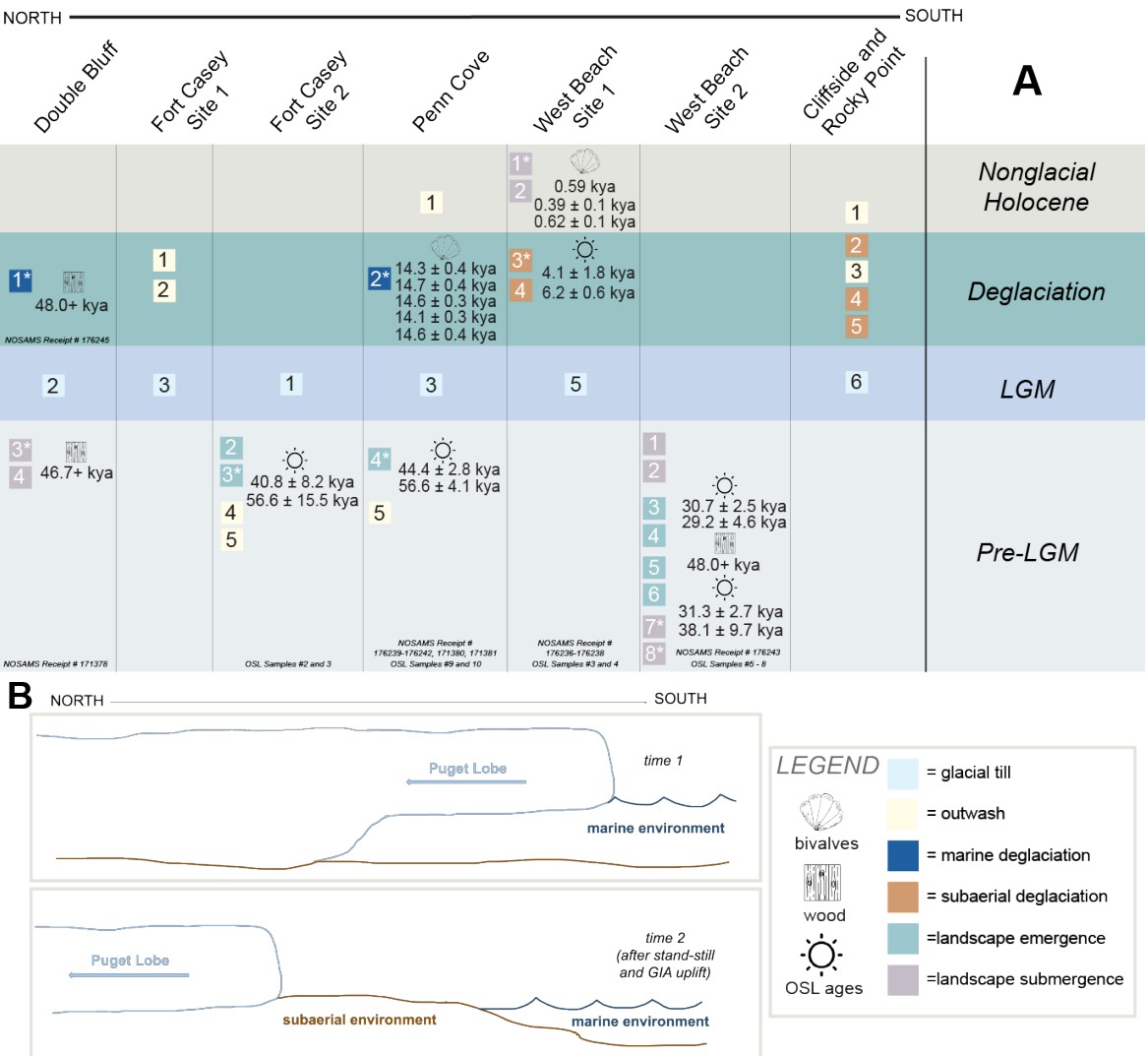

**Figure 3.** A) Grouping of facies based on depositional time periods across Whidbey Island. Units with asterisks have radiocarbon or OSL dates associated with them, shown to the right of the unit numbers. B) Top schematic drawing indicates Puget Lobe ice retreat within a marine environment at time 1. The bottom schematic showcases hypothesized northernmost retreat into a subaerial environment at time 2 after landscape reemergence from time 1. Puget Lobe ice retreat in a marine environment are only seen to have occurred at southernmost sites Double Bluff and Penn Cove while ice retreat into a subaerial environment are proposed for the West Beach and Cliffside sites.

## 4.2 Pre-LGM landscape evolution

Prior to Puget Lobe glacial advance across Whidbey Island during the LGM, several submergence and emergence facies transitions record dynamic landscape changes. Landscape emergence above sea level prior to LGM glaciation is recorded by outcrops exposed at Penn Cove and Fort Casey Site 2 (Figures 2, 3A). Penn Cove OSL ages identify this landscape emergence to occur between 56.6 ± 4.1 and 44.4 ± 2.8 kya.

Similar Fort Casey Site 2 OSL ages constrain this transition to having occurred from 56.6 ± 15.5 to 40.8 ± 8.2 kya, placing the emergence within the MIS 4 glacial and MIS 3 interglacial stages, which may be connected to a lack of ice coverage and reduced CIS loading of the solid Earth at these times.

A sequence of submergent and emergent facies are also observed in the pre-LGM deposits at West Beach Site 2. OSL dates place a submergence event between 38.1 ± 9.7 and 31.3 ± 2.65 kya while OSL dates from overlying facies places subsequent emergence between 30.7 ± 2.5 and 29.2 ± 4.6 kya (Figure 3A). Both of these events occurred within the MIS 3 interglacial. This rapid transition between landscape submergence and emergence not only identifies high sedimentation rates at this site, but also suggests that the Puget Lowland experienced rapid landscape changes during MIS 3. Clay and sand deposits included as part of the emergence and submergence interpretation may have previously been identified and referred to as the well-known Lawton Clay (Mullineaux et al., 1965) and Esperance Sands (Mullineaux et al., 1965; Crandell et al., 1966; Easterbrook, 1969; Clague, 1976; Booth, 1994). The pre-LGM timing of proglacial lake basin development, responsible for the deposition of the Lawton Clay (Mullineaux et al., 1965) and subsequent channel deposition of the Esperance Sands, has consistently been radiocarbon dated to 18,000-20,000 years ago (Mullineaux et al., 1965; Crandell et al., 1966; Easterbrook, 1969; Clague, 1976; Booth, 1994). It is highly likely that the uncertainties from our OSL-dates would contribute to discrepancy with the well-established radiocarbon dates of the Esperance Sands (Text S1; Easterbrook, 1969). Despite these uncertainties, our OSL ages provide enhanced understanding of sediment deposition and landscape evolution rates.

**4.3 LGM glacial advance**
Erosional contacts between till (Vashon Till) and underlying facies mark LGM advance of the Puget Lobe into the region. These erosional contacts beneath till are seen at multiple sites across Whidbey Island including Double Bluff, Fort Casey Site 2, and Penn Cove (Figures 2, 3B). OSL ages we collect from below the erosional contact of LGM tills (56.6 ± 4.1 and 44.4 ± 2.8 kya, within the timeframe of MIS 3) cannot be used to find precise maximum age of ice extent. However, previously radiocarbon-dated wood material place final LGM advance into the region after 17,500 cal. yr. BP (Mullineaux et al., 1965; Porter & Swanson, 1998). This major difference in ages suggests a great deal of erosion at the boundary between underlying sediments and till deposition of the Puget Lobe during ice advance. The erosional surface of outwash deposits underlying LGM till, in some areas up to 75 m (Easterbrook, 1992), is thought in part to be due to glacial scour during ice advance (Bretz 1913; Easterbrook, 1968; 1992).

**4.4 Deglaciation**

Glaciomarine sediments (Everson Glaciomarine Drift) in the uppermost 50 cm of Double Bluff Unit 1 record retreat of the Puget Lobe within a marine environment (Figure 3B; Thorson, 1980; Dethier et al., 1995; Demet et al., 2019). At Penn Cove, the presence of articulated shells in growth position (Figure 2) within a unit increasingly lacking small grain sizes upcore suggests ice retreat in a marine environment with possible tidal influence (Smith et al., 2019). With the newly developed regional MRC of 264 [14]C years, the six articulated shells found at Penn Cove were radiocarbon dated to a range of dates between $14.8 \pm 0.3$ and $14.1 \pm 0.3$ kya cal. yr. BP (Table 1). These ages strongly agree with previous literature marking marine incursion beneath the ice (Figure 1). Based on the $2\sigma$ error in glaciomarine radiocarbon dates (both those presented here and those recalculated from the literature using Marine20 and our MRC; Table 1), glacial ice appears to have been stable at Penn Cove for at least 500 years with concordant high sedimentation rates, accumulating 2.5 m during a near 700 year period. The oldest known presence of Everson Glaciomarine Drift was presented in prior work, dated to ~16,275 and 15,750 calendar years BP (Easterbook, 1992; Dethier et al., 1995; Stuiver et al., 1998; Swanson & Caffee, 2001). However, the specific location, stratigraphic context, and proximity between shells within sedimentological data are missing from these reports. The shells presented in this work were collected from within two units spanning three meters vertically within the outcrop (Figure 2) and may represent a younger limit of Everson Glaciomarine Drift deposition before final ice retreat.

Deglacial facies seen at the more northern West Beach Site 1 and Cliffside indicate ice retreat within a subaerial environment (Figure 3B). The change in ice retreat style seen from the more southern Double Bluff and Penn Cove sites to the northern West Beach and Cliffside sites may be due to the substantial stand-still of ice at Penn Cove. The presence of a GZW at Penn Cove (Figure 1; Simkins et al., 2018; Demet et al., 2019) further supports the idea that the marine-terminating ice was pinned at this location for a substantial amount of time. Additionally, the Rocky Point site features a bedrock high (i.e., a potential pinning point of ice; Hogan et al., 2020), and other mapped GZWs (Figure 1) suggest several points across Whidbey Island could have periodically stabilized ice due to land rebound during final ice retreat (Simkins et al., 2018; Demet et al., 2019). The paired stratigraphic, geomorphic, and geochronological-based evidence of period ice stability on Whidbey Island provides empirical evidence for sedimentation and GIA as possible mechanisms for ice stabilization during retreat. Within this work, we also identify that the Puget Lobe experienced stepwise retreat, rather than catastrophic loss of ice due to rapid unpinning (c.f., Easterbrook, 1992).

**4.5 Nonglacial Holocene landscape evolution**

In the sediment record following final glacial infuence, the Penn Cove and Cliffside sites contain outwash deposits from proglacial fluvial sources. An OSL age within the submergence facies of Unit 2 at West Beach Site 1 marks the subsequent

transition from a post-glacial fluvial environment to a submarine environment between 6.2 ± 0.6 and 4.1 ± 1.8 kya (Figures 2, 3A). Radiocarbon-dated shell hash sampled from the uppermost unit at this same West Beach Site 1 suggests a highly energetic aquatic marine or coastal environment was present in this location as early as 0.62 ± 0.1 kya cal. BP through at least 0.39 ± 0.1 kya cal. BP (Figure 2, 3A). After a slow in initial lithospheric rebound from ice-loading or a possible local tectonic event, it is feasible vertical land movement slowed enough to allow local sea level to resubmerge the region between 600 and 390 years ago (Figure 2, 3A). In our analysis of nonglacial Holocene sediments in the Puget Lowland, we find that this landscape still evolves rapidly due to ongoing affects of GIA and tectonics in the region.

**5 Conclusions**

This decimeter-scale physical sedimentological assessment, paired with geochronological assessment of seven sites across the deglaciated Puget Lowland, provides spatiotemporal information on landscape emergence and submergence as well as final ice advance and retreat of the southernmost CIS. Rates of vertical landscape changes constrained through OSL dating indicates the Puget Lowland was a highly dynamic region where a sequence of landscape emergence and submergence occurred within ~1,000 years during MIS 3. This work develops a local MRC applied to new and existing glaciomarine radiocarbon dates, all of which agree that timing of marine inflow to the grounding line occurred between 15,000 and 14,000 years BP. Radiocarbon dates paired with sedimentology and existing geomorphology show at least 500 years of ice marginal stand-still and substantial grounding zone sedimentation during final ice retreat. We show the Puget Lobe experienced stepwise retreat with GIA and grounding-line sedimentation as possible mechanisms for stabilization of the ice margin. While more southern sites (e.g., Double Bluff and Penn Cove) record ice retreat within submarine environments, the northernmost sites (e.g., Cliffside and Rocky Point), appear to record ice retreat into a subaerial environment. The similarities between the rheology in this location and the rheology of the Antarctic Peninsula, as well as the topographic similarities between the Puget Lowland and modern margins of the Greenland Ice Sheet make these findings highly relevant to increasing process-based understanding of solid Earth influence on ice dynamics in contemporary marine-terminating glacial systems.

**Acknowledgments**

The sites analyzed for this work are located on land historically cultivated and inhabited by the Skokomish, Suquamish, Squaxin, Stl'pulmsh, Steilacoom, Puyallup, Muckleshoot, and Duwamish peoples, while much of the data analysis and interpretation were conducted on land cultivated and inhabited by the Monacan Nation. The peoples of these Nations were custodians of the land for time immemorial before forced removal and genocide during colonization. The authors acknowledge their ongoing stewardship of the lands.

The authors would like to thank three anonymous reviewers and R. Venturelli for their constructive and thoughtful comments on this work. The manuscript was greatly improved because of their contributions and insights. This work was funded by the

Chamberlain Endowment and the H.G. Goodell Endowment at the University of Virginia. The funding and support for the radiocarbon dates presented was made possible through the NOSAMS Graduate Student Internship at Woods Hole Institute, supported by NSF cooperative agreement OCE-1755125, and the Burke Institute. Thank you to Dr. Mark Kurz, Dr. Roberta Hansman, Anne Cruz, Mary Lardie Gaylord, and Nan Trowbridge for

their hospitality and guidance throughout M. McKenzie's internship. The authors declare that they have no conflict of interest.

**Open Research**

Digital data including site coordinates and sample grain size, trace element (not included

in analysis), moisture content, and magnetic susceptibility data and all 236 physical samples are housed in the PANGAEA database (McKenzie et al., 2024) and at the Washington Department of Natural Resources at the Washington Geological Survey. Physical samples are in WhirlPak bags, labelled by site name, number, and sampling interval in centimeters. When collected in the field, unit names were given from down-to-

up outcrop. For the purpose of simplicity, the unit names were flipped for manuscript analyses to be listed as smallest to highest up-to-down outcrop. To request physical data, please contact Jessica Czajkowski (Jessica.Czajkowski@dnr.wa.gov) and/or Ashley Cabibbo (Ashley.Cabibbo@dnr.wa.gov) at the Washington State Department of Natural Resources.

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
