# Peer review of "Spatial variability of marine-terminating ice sheet retreat in the Puget Lowland"

_EGUsphere, 2023_

## Author Response (AR1)

Dear Editor Atle Nesje and Reviewers,

Thank you all very much for your comments and suggestions on the manuscript, "Spatial variability of marine-terminating ice sheet retreat in the Puget Lowland". We hope you agree that the incorporation of your thoughtful edits greatly improved the manuscript and its readability. Below, I will share some of the larger edits made to this work based on themes from all three reviewer comments, followed by line edits addressing comments.

General structure and analysis comments:
- Probably most importantly to these edits, I realized during the revision process that I had recalibrated my radiocarbon ages to calendar year BC rather than BP. Upon proper recalibration, I've found the radiocarbon ages we collected much more closely match those within the literature (Table 1). While our claimed details on timing of retreat and our dates' relationship to the published literature changed significantly, many of our base findings remain consistent. We believe the evidence we provide sufficiently exemplifies that retreat across the Puget Lowland occurred in a stepwise fashion, with a transition between submarine and subaerial deglaciation from south to north.
- Within the "Regional Context" section of the introduction, I have added two subsections titled "Pre-Fraser glaciation and Fraser glaciation deposits" and "Deglacial and nonglacial Holocene deposits" that includes much of the text that was originally in the supplement. In creating this section, I expanded upon the supplement text by providing varying viewpoints from the literature and including perspectives on local geomorphology within each section. In this section, I have also highlighted gaps and shortcomings of previous research that leave room for the reanalysis we provide.
- In the interpretation, I now incorporate much of the language and references from the "Regional Context" section to provide continuity between what is established in the literature and the new information we provide.
- The newly developed Figure 1 incorporates glaciomarine shell radiocarbon dates from Swanson & Caffee (2001), Dethier and others (1995), and select ages from Easterbrook (1992) marking start of ice retreat in the Puget Lowland. The ages selected for incorporation were provided in $^{14}$C years, so in order to make them comparable to the ages presented in this work, I recalibrated the ages using Marine20 and our marine reservoir correction of 264 $^{14}$C years. This figure now also includes proposed retreat sequences from Ehlers and others (2010) for the southern Cordilleran at 13, 14, and 15 ky BP. Another panel in the updated figure includes a map of Whidbey Island with topographic hillshade to highlight the presence of streamlined subglacial bedforms. Outcrop sites are shown on this map in addition to the inferred and identified grounding zone wedges from Demet and others (2019). I believe this new standalone figure addresses many of the comments received about a lack of context in the previous regional figure.

- The updated Figure 2 addresses many of the provided comments including making text more readable than the original figure, adding a legend for stratigraphic symbols, including a scale for grainsize transitions in the graphs, and incorporating pictures of the outcrops. The pictures of the outcrops were generally chosen based on accessibility, visibility of sediment structures, and inclusion of radiocarbon and OSL sample collection. The color scheme of the grain size and unit classification was kept the same, but boxes around each site and an explicit statement of noncorrelation between sites should be sufficient to avoid confusion. A clarification on the white dots was provided in the methods and the figure caption to improve understanding of the difference between collection locations. The magnetic susceptibility (MS) information was moved to the supplement. Both the MS figure and Figure 2 now include a statement in the figure caption about sampling resolution changing as a function of accessibility. While still dense with information, the Figure 2 text size was increased, and the panels were rearranged to maximize visibility.
- Figure 3 now includes the geochronology ages on the main panel with sample reference numbers at the bottom of each column. The colors were improved to be more distinct from one another and a legend provides details on the symbols used. The term "deglacial" was changed to "nonglacial Holocene" in this figure and changed to "nonglacial Holocene landscape evolution" in the interpretation and discussion. The schematic from the supplement was added to this figure and now depicts southern retreat into a marine environment vs. a northern retreat in a subaerial environment as suggested.
- Table 1 – the table of radiocarbon ages – now includes new data from this work and data from previous work (Easterbrook, 1992; Dethier et al., 1995; Swanson & Caffee, 2001). I have added columns for "Sample location (site, unit)", changed the original column for sample name to "Laboratory number", reassessed the "Type" column, changed error of the cal. year BP ages to 2σ, added columns for latitude and longitudinal coordinates, and incorporated a column for the age reference. The ages used to calculate the marine reservoir correction (MRC) were placed into the supplement. In the MRC radiocarbon table, the column for "Actual age" was changed to "Time since live-collection (cal. years BP).
- The limitation of the developed MRC, as mentioned by several reviewers, was explicitly stated in the introduction and the methods. I have kept the MRC development as part of the methods, however, because the MRC was used to further develop many of the results. While reservoir effects vary through the Pleistocene (Schmuck et al., 2021), the development of this local correction, on top of the Marine20 calibration -- in my opinion -- will more accurately place ages than not using an MRC at all. Therefore, I applied the MRC to my older dates but did explicitly state the uncertainties in doing so, however it is the most local MRC to date and therefore carries some merit in that regard.

- In response to a comment focused on the suitability of the title, based on the support we provide, we have decided to retitle the manuscript "Spatial variability of marine-terminating ice sheet retreat in the Puget Lowland".

Specific line comments [line comments are made in reference to the original manuscript. The line numbers have since changed]:
- The large overdispersion in the Fort Casey Site 1 OSL age made it highly unreliable and was therefore removed from the analysis. Other samples that still had >20% overdispersion but could still be interpreted were included in the analysis (West Beach and Penn Cove – OSL1). Information about overdispersion was added as a column to Table 2 and included as a disclaimer in the results section reporting the ages. Total dose rate remains in Table 2, but all other dose and dose rate information was developed into Tables S4 and S5 in the supplement, respectively. Additional edits were made to Supplemental Text 1 in regards to overdispersion and explanation of OSL age calculations.
- In reference to Swanson & Caffee (2001) in the introduction, any mention of their cosmogenic nuclide production rates was referenced properly, rather than referring to their contributions as "cosmogenic exposure ages".
- Lines 71-74 and 572-573: the references of Nield et al. (2014) and Whitehouse et al. (2019) were adjusted within the sentence to reference the part in which each study supports.
- Clarifications were provided regarding why I draw connections between Greenland's outlet glaciers and the Puget Lowland. In this, I reference the previous work of Eyles et al. (2018) who directly compares these two glacial systems through subglacial topographic conditions.
- Lines 66-67: The GIA acronym error was corrected.
- Line 100: Text was changed from "magnitude of landscape emergence (cm/a)" to "rate".
- Line 141: The typo of "grain" was corrected here.
- Line 156: Text was changed to "Sediment samples were collected from […]".
- Line 160: Text was changed to "[…] referred to as lenses […]".
- Line 182: This citation was changed from Goehring et al., 2019 to NOSAMS, 2023.
- Lines 196-197: This sentence was reworded for greater clarity and the term "live collected" is now used throughout the manuscript.
- Line 211: The extra "and" was removed from this sentence.
- Line 221: "Materials" was changed to "sediments".
- Line 256: The typo of "considerably" was corrected.
- Lines 262-264: A reference to Figure 2 was added to this line. This line was also reworded for clarity.

- Lines 332-333 and 428-430: Here, I clarify West Beach Unit 3 consists of gravel with parallel-to-bedding clast orientation and include this clarification in the classification of subaerial glacial retreat facies.
- Line 388: I changed the word "structure" to "facies" in this sentence.
- Lines 397-398: References for glacial till and outwash were improved by including a recent textbook from Hagg (2022) and the Evans (2006) mentioned in Reviewer 3's comment.
- Line 410: Here and in all references to the word "winnowing" I shift to using a descriptor of "absence of fines" and refer to a *process* of winnowing via tidal flexure (Smith et al., 2019).
- Line 413: The highlighted typos were corrected.
- Line 433: I removed the word "respectively" to make this sentence more clear.
- Line 434-435: I removed the reference to glacial isostatic adjustment as a driver of sea level in this specific instance.
- Line 439-442: This line was edited with Reviewer 3's recommended edits.
- Line 442-445: I rearranged this sentence to make sure each reference was used within-range.
- Line 501: The error of "MIS 5" was changed to "MIS 3" in this sentence.
- Line 508: All instances of the word "glacimarine" such as in this line, were changed to "glaciomarine".
- Line 515-518: Reevaluation of our ages resolved this comment; however, I did add some text in this section about the shells in "growth position" in an area with little slope (unlikely for a slump).
- The title of section 4 was changed to "Interpretation and Discussion".
- I have developed a separate citations list for the supplement. The manuscript and the supplement now stand independently in terms of the references.
- All occurrences of "glacial till" were changed to simply "till".
- The use of dashes throughout the manuscript was remedied and should all be consistent (with much fewer dashes) now. The text size, font, and spacing were standardized across the document.

Thank you again for your comments, we believe this manuscript is much stronger and are confident it is ready for next stages in this review process.

Best,
Marion McKenzie and co-authors